

# Multi-scale relationship between land use/land cover types and water quality in different pollution source areas in Fuxian Lake Basin

Shihua Li[1], Shuangyun Peng[2], Baoxuan Jin[3], Junsong Zhou[1] and YingXin Li[2]

[1] Yunnan Provincial Geomatics Centre, Kunming, Yunnan, China
[2] College of Tourism & Geographic Sciences, Yunnan Normal University, Kunming, Yunnan, China
[3] Information Center, Department of Natural Resources of Yunnan Province, Kunming, Yunnan, China

Corresponding author
Shuangyun Peng, frankmei@126.com

## ABSTRACT

The spatial-temporal evolution of land use and land cover (LULC) and its multi-scale impact on the water environment is becoming highly significant in the LULC research field. The current research results show that the more significant scale impact on LULC and water quality in the whole basin and the riparian buffer scale is unclear. A consensus has not been reached about the optimal spatial scale problem in the relationship between the LULC and water quality. The typical lake basin of the Fuxian Lake watershed was used as the research area and the scale relationship between the LULC and water quality was taken as the research object. High resolution remote sensing images, archival resources of surveying, mapping and geographic information, and the monitoring data of water quality were utilized as the main data sources. Remote sensing and Geometric Information Technology were applied. A multi-scale object random forest algorithm (MSORF) was used to raise the classification accuracy of the high resolution remote sensing images from 2005 to 2017 in the basin and the multi-scale relationship between the two was discussed using the Pearson correlation analysis method. From 2005 to 2017, the water quality indicators (Chemical Oxygen Demand (COD), Total Phosphorous (TP), Total Nitrogen (TN)) of nine rivers in the lake's basin and the Fuxian Lake center were used as response variables and the LULC type in the basin was interpreted as the explanation variable. The stepwise selection method was used to establish a relationship model for the water quality of the water entering the lake and the significance of the LULC type was established at $p < 0.05$. The results show that in the seven spatial scales, including the whole watershed, sub-basin, and the riparian buffer zone (100 m, 300 m, 500 m, 700 m, and 1,000 m): (1) whether it is in the whole basin or buffer zone of different pollution source areas, impervious surface area (ISA), or other land and is positively correlated with the water quality and promotes it; (2) forestry and grass cover is another important factor and is negatively correlated with water quality; (3) cropping land is not a major factor explaining the decline in water quality; (4) the 300 m buffer zone of the river is the strongest spatial scale for the LULC type to affect the Chemical Oxygen Demand (COD). Reasonable planning for the proportion of land types in the riparian zone and control over the development of urban land in the river basin is necessary for the improvement of the urban river water quality. Some studies have found that the relationship between LULC and water quality in the 100 m buffer zone is more significant than the whole basin scale. While our study is consistent with

the results of research conducted by relevant scholars in Aibi Lake in Xinjiang, and Erhai and Fuxian Lakes in Yunnan. Thus, it may be inferred that for the plateau lake basin, the 300 m riparian buffer is the strongest spatial scale for the LULC type to affect COD.

# INTRODUCTION

With the rapid development of the global society and economy, greater attention has been paid by the international academic community to research the effects that LULC has on the water environment by *Johnson et al. (1997)*, *Meneses et al. (2015)*, *Sajikumar & Remya (2015)*, *Zhou et al. (2016)*. The quality of the water environment of the basin is an important foundation for the harmonious development of hydrology-ecology-economics in the basin and unreasonable LULC changes are one of the vital characteristics that affect regional changes in the water environment (*Zhang, Wang & Li, 2003*; *Thomas, Bond & Hiscock, 2013*). At present, the research on LULC type and water quality focuses on the response of water quality indicators such as TN (Total Nitrogen), TP (Total Phosphorous), pH (hydrogen ion concentration), CODMn (Chemical Oxygen Demand-Mn), TSS (Total Soluble Solid), BOD5 (Biochemical Oxygen Demand, BOD), DO (Dissolved Oxygen), etc. to LULC changes from land use, land use structures, and land use patterns and has reached a consensus. There was a significant negative correlation between forest land, grassland, and water quality indicators, while arable land and construction land showed a strong positive correlation (*Zhang, Cheng & Xiang, 2011*; *Yang et al., 2017*). However, due to the multi-scale and distribution pattern of land use (*Tu, 2011*; *Zhou et al., 2012*), there is a significant scale correlation between LULC changes and water quality indicators, which leads to uncertainty about the relationship between land use patterns and river water quality. From the saliency of spatial influence, the sub-basin scale is significantly higher than the riparian buffer scale (*Sliva & Williams, 2001*; *Jarvie, Oguchi & Neal, 2002*; *Woli et al., 2004*; *Li, Xu & Li, 2012*), and some studies have reached the opposite conclusion (*Sahu & Gu, 2009*; *Li et al., 2009*; *Huang et al., 2011*; *Ou, Wang & Geng, 2012*). The current research results show that the more significant scale impact on the land use and water quality in the whole basin and riparian buffer scale is unclear. The optimal or strongest spatial scale problem of the relationship between land use and water quality has not reached a unanimous conclusion (*Johnson et al., 1997*; *Sliva & Williams, 2001*; *Shen et al., 2015*; *Ding et al., 2016*).

Fuxian Lake is a unique low-latitude and high-altitude plateau lake ecosystem. It is an important international lake in which to study the mechanism of biodiversity formation. Affected by the East Asian and Southwest monsoons, it is the most sensitive representative lake in response to global change and has become one of the hot spots of the research on international lakes, favored by experts and scholars at home and abroad. It is one of the plateau lake basin systems valued by researchers for having one of the most ecologically fragile areas in geoscience in China. A literature review on Lake Fuxian reveals that many

scholars have carried out research on its hydrological and water qualities (*Xia, Li & Xiong, 2007*; *Pan et al., 2008*; *Gao et al., 2013*; *Zhai et al., 2015*; *Zhang et al., 2015*; *Yan et al., 2016*; *Yao et al., 2017*; *Chen et al., 2019*), aquatic organisms (*Xiong et al., 2006*; *Li et al., 2007*; *Li et al., 2017a*; *Li et al., 2017b*), eutrophication (*Li et al., 2003*; *Zhang et al., 2012*; *Xu et al., 2016*), LUCC (*Liu, Wu & Gao, 2008*; *Li et al., 2015*; *Yang et al., 2015*; *Yang et al., 2016*; *Dai et al., 2017*; *Li et al., 2017a*; *Li et al., 2017b*), soil erosion and land degradation (*Yang et al., 2016*; *Ma et al., 2016*), vegetation change (*Li et al., 2016*), water environment economy (*Xia et al., 2010*; *Gao, Chen & Guo, 2014*; *Xiong et al., 2006*), and sustainable development (*Duan et al., 2013*) based on remote sensing, field monitoring, computer simulation, laboratory analysis, and other technologies. Their results show that, in recent years, with an increase in global changes, a higher level of urbanization, and the acceleration of social and economic development in the basin, the body of Fuxian Lake has shrunken, the water level has decreased significantly, and the water area has gradually narrowed. The LULC changes in the basin are significant, soil erosion and land degradation are serious, and the ecological environmental quality of the basin is generally declining (*Yang et al., 2015*; *Yang et al., 2016*; *Ma et al., 2016*; *Li et al., 2017a*; *Li et al., 2017b*). The water quality of the river entering the lake and the lake shore are severely polluted. Thus, the water quality of the lake is seriously threatened. Therefore, clarifying the scale relationship between the land use and water quality will lay a foundation for the improvement and protection of the water quality in the basin. Our goal in this paper was to investigate the historical shifts in LULC in Lake Fuxian between 2005 and 2017. We also examine the spatial scales at which these changes impact water quality by monitoring the water quality data from the local department governing Lake Fuxian throughout the watershed, together with data from five land cover maps (2005–2017). The present study addresses the following research questions: (1) What land use category has the strongest effect on water quality? (2) How do spatial and temporal variations in LULC within and across watersheds influence water quality metrics in the Lake Fuxian watershed? (3) At what spatial scale does the LULC type act to influence water quality? (4) How the multi-scale relationship between land use types and water quality indicators can be used to balance relationship between land use and water protection?

## MATERIALS & METHODS

### General situation of the study region

Fuxian Lake is located in the center of the Central Yunnan Basin, Yuxi City, Central Yunnan Province, China. It is China's largest deep-water freshwater lake, the first large lake as the source of the Pearl River, and part of the Nanpan River system. Its geographical location is 24°21′28″–24°38′00″N and 102°49′12″–102°57′26″E (Fig. 1). As one of the nine plateau lakes in the Yunnan Province, Fuxian Lake was the second deep-water lake explored in China and the lake area and its water storage amount to the 8th and 3rd in China, respectively (*Wang & Dou, 1998*). Because of the particularity of its geographical location, its powerful water supply capacity, and its recreational value it is known as the "plateau pearl" in Central Yunnan. It serves as an important and reliable resource for

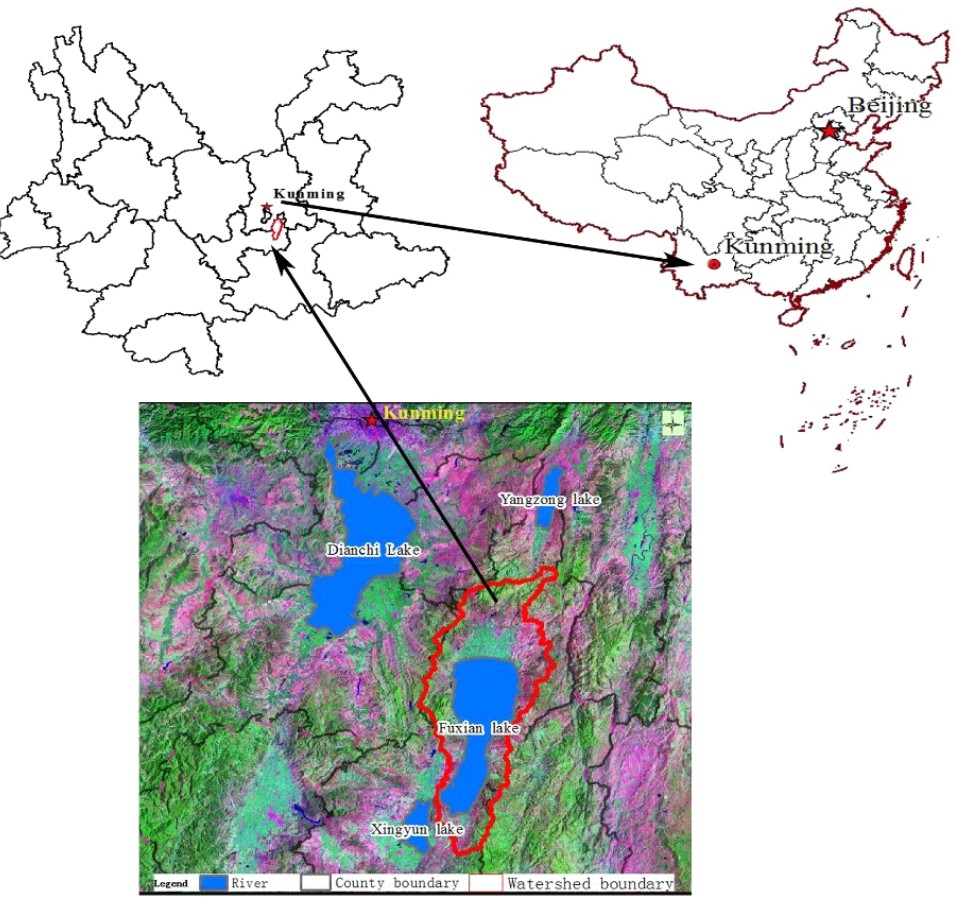

**Figure 1** Location of Fuxian lake watershed.

the sustainable social and economic development in Central Yunnan, the strategic water resource for the regional development of the Pan-Pearl River Delta, and an important strategic source of drinking water in the Pearl River Basin and Southwest China (*Gao et al., 2013*). The water quality of Fuxian Lake is Class I and it is one of the best natural lakes in China. The vegetation in the basin is mainly secondary vegetation such as grass, shrub, and coniferous forest and the population reaches about 160,300. The rural economy is dependent upon crop production, and the main food crops include rice, corn, and wheat and economic crops include flue-cured tobacco and rape. Industry is dominated by the phosphorus chemical industry, building materials, food processing, and aquatic products of which the phosphorus chemical industry is the pillar industry of this area. The land use type of the Fuxian Lake Basin has always been dominated by forests and water areas but with the improvement of the urbanization level of the basin, human activities have increasingly disturbed the natural environment, and phenomena such as the reclamation of lakes, deforestation, over-exploitation of tourism resources, and the rapid increase of functional buildings have resulted in a significant change in the land coverage types in the Fuxian Lake Basin.

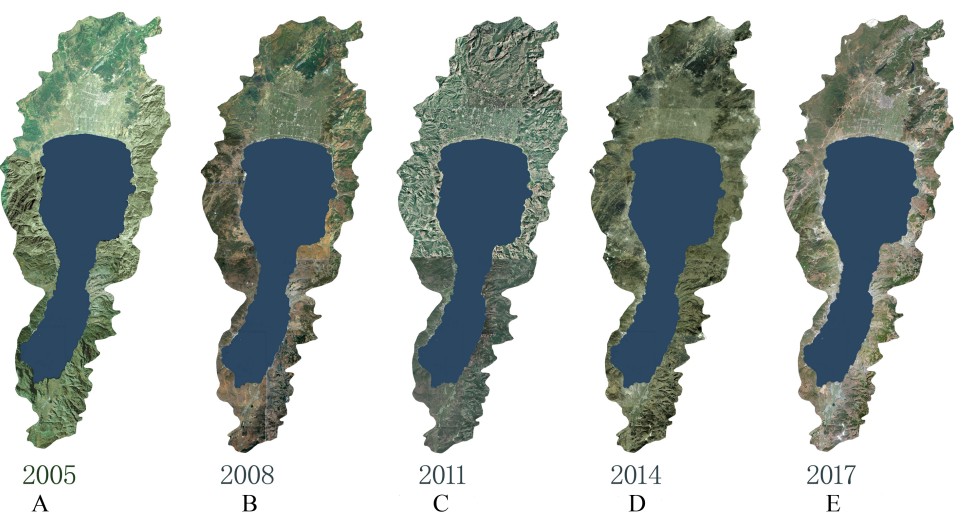

2005     2008     2011     2014     2017
A       B       C       D       E

**Figure 2**   **Remote sensing image of study area.** (A) 2005, (B) 2008, (C) 2011, (D) 2014, (E) 2017.

## Data sources

Because of the obvious seasonal variation of rainfall in the Fuxian Lake Basin and the large
seasonal variation of the lake area, the remote sensing images of relatively stable lake water
levels during the dry season from January to March were selected. These include the five
phases of high spatial resolution remote sensing image data in January 2008 (QuickBird
satellite data, 0.61 m full color and 2.44 m multi-spectrum), January 2011 (WorldView-2
satellite data, 0.5 m full color and 1.8 m multi-spectrum), January 2014 (WorldView-2
satellite data, 0.5 m full color and 1.8 m multi-spectrum), and March 2017 (Beijing No.
2 satellite data, 0.5 m full color and 1.8 m multi-spectrum) (Fig. 2). Data was purchased
from remote sensing image agents.

The reference data sources of orthorectified remote sensing images include 1:10000
DLG, 0.5m resolution DOM of 1:25000, DEM data of 10 m grid spacing of the basin,
1:10000 out-of-flight control results, the three-space encryption results, 1:10000 DOM
data results in the research area measured from 1985 to 2013, as well as achievements
of the set basic GPS C-level points, triangle points of each level, military control points
and standard points. The data are from the Yunnan Provincial Bureau of Surveying and
Mapping. The training data and test data extracted by LULC adopts the WorldView-2
remote sensing image with a resolution of 0.5 m in 2012 from the results of the first
national geographical situation survey in the Yunnan Province and surface coverage
classification data (water area, desert and bare surface, construction land, arable land,
garden land, forest land, structure, grassland, road, and house building area), which were
verified and modified in the field in 2014. The data are from the Yunnan Provincial Bureau
of Surveying and Mapping. The auxiliary data for land use change driving force analysis
and suitability atlas production includes the database of results from the first (1996) and
the second (2006) national land survey of the study area, agricultural land grading results,
and data from the land use planning database (updated in 2005 and other years), with data
from Yuxi Municipal Bureau of Land and Resources. The 13th Five-Year Plan of Fuxian

Lake Basin (2016-2020), and the data of Fuxian Lake Basin Planning are from the Fuxian Lake Administration Bureau of Yuxi City. The annual average rainfall, annual average temperature, economy and population data of the basin are obtained from the Statistical Yearbook of Yuxi City. The water quality indicators were typically collected monthly TN, TP, and CODmn with the highest pollution load in the main rivers and lakes in 2005, 2008, 2011, 2014, and 2017 and the three indicators were obtained from monthly reports on the quality of the surface water provided by the Environmental Monitoring Center of Chengjiang County.

## LULC information extraction

The first-class classification system (water area, desert and bare surface, arable land, structure, garden land, grassland, forest land, house building, road and construction land) in the general survey contents and indicators of geographical conditions (GDPJ 01-2013) was adopted as a classification system for extracting LULC information of the remote sensing images. The definition of each type is shown in GDPJ 01-2013. Due to the differences between remote sensing image data sources and the time phase, training samples and test samples were collected from the remote sensing images in 2005, 2008, 2011, 2014, and 2017, respectively. For each LULC type of images in each phase, 2000 samples were collected, 70% of which were used as training samples and 30% as test samples. The multi-scale object random forest (MSORF) algorithm (refer to *Li, Li & Shao, 2016* for details) was used to obtain the LULC classification results of the river basin in 2005, 2008, 2011, 2014, and 2017. The classification accuracy Kappa is generally 0.8 or more, (i.e., 2005 (0.835), 2008 (0.812), 2011 (0.819), 2014 (0.822), and 2017 (0.805)). Referring to the current land use map and remote sensing images, the classified data are manually modified to form the final LULC information (Fig. 3). To highlight the differences in the effects of the natural surface cover and human surface cover types on water quality, the LULC types were classified and combined according to the LULC type classification criteria. i.e., arable land and garden land merged into cropping land (CL), woodland and grassland merge into forestry and grass cover (FGL), buildings (districts), roads and structures were combined into an impervious surface area (ISA), the artificially excavated land, desert, and exposed surface were merged into other land (Other), and the water area was unchanged. The correlation analysis with the water quality of the rivers entering the lake was carried out at different spatiotemporal scales based on the combined LULC types.

## Scale definition

Scale selection is related to the experimental design and the information collection in scale research, which is the starting point and foundation of our research. The selection of different scales will affect the comprehension of ecological patterns, processes, and their interactions to a certain extent and ultimately affect the scientific and practical research results (*Lv & Fu, 2001*; *Wu, Liu & Wang, 2003*; *Zhu, Tian & Zhang, 2005*). The discussion of the scale relationship between LULC and water quality is mainly focused on the basin and riparian buffer scales. It is generally believed that the 90 m, 100 m, and 300 m riparian buffers are sensitive scales, while others consider that the whole basin scale is a sensitive scale.

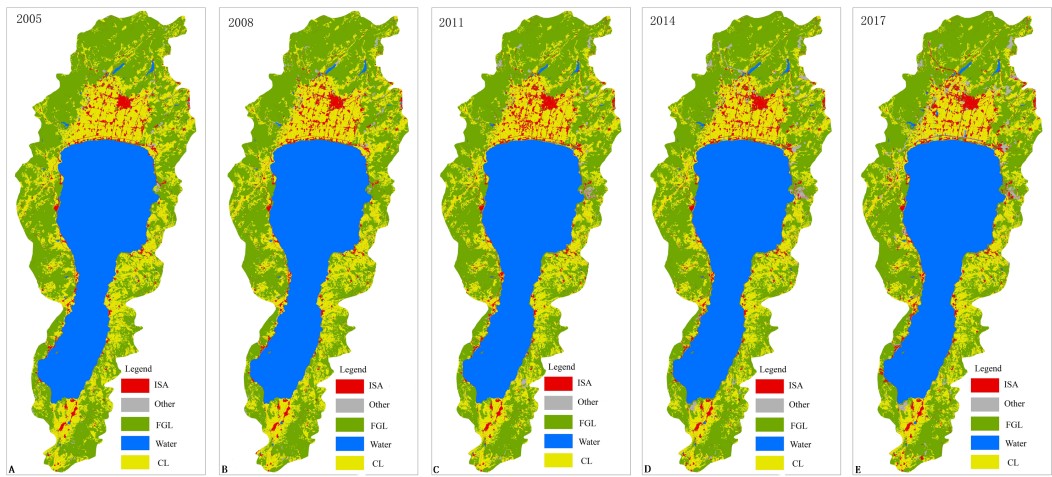

**Figure 3** **LULC type of study area.** (A) 2005, (B) 2008, (C) 2011, (D) 2014, (E) 2017.

Three representative rivers that have been polluted by phosphate mines, urban development, and village farmland were selected as the analysis objects after considering factors such as the composition of the pollution source of the river basin, the topography and geomorphology, and the LULC distribution. The Maliao River (MLH), which is seriously polluted by the county town, the Daicun River (DCH) and the Dongda River (DDH) polluted by phosphate mines and phosphorus chemical enterprises, and the Liangwang River (LWH), Shanchong River (SCH), Jianshan River (JSH), Niu Mo River (NMH), Luju River (LJH) and Ge He River (GH) are seriously polluted by the farmland in the nearby village. In order to facilitate comparison with relevant research results, the response relationship between the water quality of the nine rivers entering the lake in the above three types of regions and the corresponding sub-basin scale and buffer-scale LULC type were analyzed. Based on the results of previous studies, the spatial scale for exploring the relationship between the water quality and LULC is defined as: the riparian buffer scale of the nine rivers (100 m, 300 m, 500 m, 700 m, 1,000 m), sub-basin scales (MLH, GH, LJH, SCH, DDH, DCH, LWH, JSH, NMH) and the whole basin (Fig. 4). The time scale defines the interannual scale.

The processing of the riparian buffer was done under the ArcGIS platform. Based on the topographical features of the basin, the sub-basin division was performed in the ArcSWAT tool based on the DEM with 2 m grid spacing and water of the 1:10000 DLG in the study area. Considering that the terrain of the north bank of the river basin is flat, in order to ensure that the generated sub-basin was in line with the actual situation, a water network of 1:10000 DLG was loaded into the Burn in a stream network to participate in the calculation. The 247 sub-basins generated by automatic segmentation were recombined and were merged into 28 in total. The nine main rivers entering the lake were extracted as sub-basins. The exit of the sub-basin was the entrance into the lake and the scope of each sub-basin included the entire area upstream of the outlet.

Peer

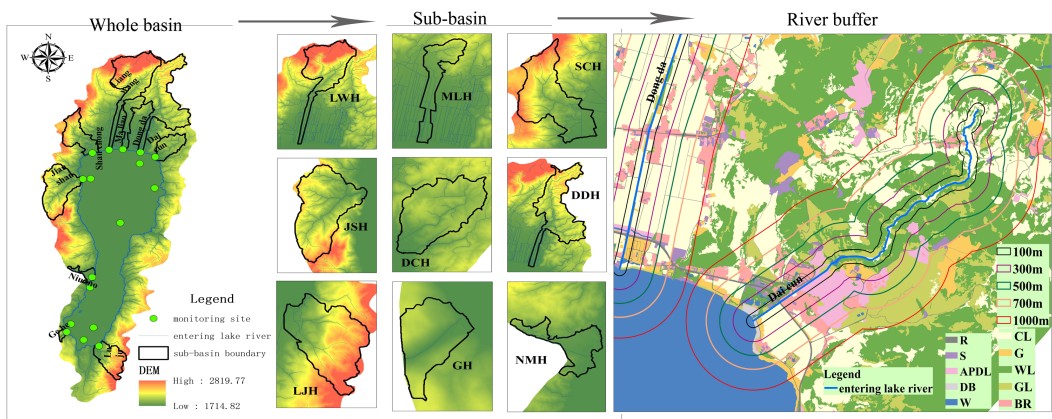

**Figure 4  Spatial scale of study.**

## Statistical analysis method

The correlation between the water quality and LULC was analyzed based on the Pearson correlation analysis method in the SPSS software. The monthly water quality indicators from 2005 to 2017 (Chemical Oxygen Demand (COD), Total Phosphorous (TP), Total Nitrogen (TN)) of the nine rivers in the lake basin and the Fuxian Lake center were used as response variables, and the LULC type in the basin was interpreted as an explanation variable. Using the stepwise entry-removal (SER) method (*Hu, 1990*) to establishing a relationship model for the water quality of the entering lake and the LULC type at $p < 0.05$. The water quality index is the dependent variable, and the LULC type is the independent variable in the model. The entry probability of the model is 0.05 and the rejection probability is 0.1.

## Water quality evolution method

The three water quality indicators of the nine rivers were analyzed according to the "Surface Water Environmental Quality Standard (GB3838-2002)" using the single index statistical method in five time periods.

## RESULTS

### Water quality characteristics of the study area

The water quality indicators COD, TN, and TP of the nine rivers entering the lake and the lake center was analyzed using descriptive statistical methods. The results revealed that the COD concentration in the rivers entering the lake ranged from 1.10 to 100.17 mg/L, and the average was 20.76 mg/L. The TP concentration ranged from 0.025 to 2.54 mg/L, and the average was 0.32 mg/L; the TN concentration ranged from 0.322 to 32.91 mg/L, and the average was 7.75 mg/L. The three indicators of the lake's quality change very little, and TP and TN were basically unchanged (Table 1).

According to the evolution of water quality it was found that among the 45 sections, the proportion of COD in the class III to the inferior class V was 47.67%, TP was 75.56%; TN

**Table 1  Lake center and river water quality statistics.**

| Site | Water quality index | Samples | Minimum (mg/L) | Maximum (mg/L) | Average (mg/L) | Standard deviation (mg/L) | Variance (mg/L) |
|---|---|---|---|---|---|---|---|
| | COD | 45 | 1.10 | 100.17 | 20.76 | 20.67 | 426.87 |
| River entering lake | TP | 45 | 0.025 | 2.54 | 0.32 | 0.43 | 0.19 |
| | TN | 45 | 0.322 | 32.91 | 7.75 | 7.18 | 51.49 |
| | COD | 5 | 1.03 | 1.44 | 1.206 | 0.195 | 0.038 |
| Lake center | TP | 5 | 0.001 | 0.007 | 0.004 | 0.002 | 8.00E−06 |
| | TN | 5 | 0.155 | 0.177 | 0.166 | 0.009 | 8.00E−05 |

**Table 2  The LULC variation characteristics in different pollution source areas (percentage, %).**

| LULC types | Scale | 2005 | 2008 | 2011 | 2014 | 2017 |
|---|---|---|---|---|---|---|
| | phosphate mine basin | 35.52 | 34.86 | 33.61 | 31.31 | 30.65 |
| cropping land | urban | 52.51 | 49.20 | 46.91 | 45.92 | 41.30 |
| | Village and farmland basin | 26.10 | 25.77 | 24.62 | 24.09 | 23.53 |
| | Whole basin | 23.96 | 23.48 | 22.47 | 21.93 | 21.63 |
| | phosphate mine basin | 57.09 | 57.03 | 58.08 | 58.54 | 56.84 |
| forestry and grass cover | urban | 29.15 | 29.21 | 29.95 | 32.19 | 31.37 |
| | Village and farmland basin | 69.01 | 68.71 | 70.10 | 70.51 | 69.74 |
| | Whole basin | 39.68 | 39.79 | 40.34 | 40.6 | 39.85 |
| | phosphate mine basin | 4.06 | 4.25 | 4.93 | 4.95 | 5.78 |
| impervious surface area | urban | 16.87 | 20.02 | 21.66 | 19.02 | 19.09 |
| | Village and farmland basin | 3.61 | 4.10 | 4.41 | 4.06 | 4.76 |
| | Whole basin | 3.35 | 3.66 | 4.13 | 3.95 | 4.4 |
| | phosphate mine basin | 1.75 | 2.32 | 2.06 | 3.73 | 5.26 |
| other land | urban | 0.20 | 0.55 | 0.73 | 1.61 | 6.78 |
| | Village and farmland basin | 0.10 | 0.42 | 0.13 | 0.26 | 0.75 |
| | Whole basin | 0.48 | 0.62 | 0.82 | 1.19 | 1.68 |
| | phosphate mine basin | 1.57 | 1.54 | 1.33 | 1.47 | 1.47 |
| water | urban | 1.28 | 1.01 | 0.75 | 1.26 | 1.46 |
| | Village and farmland basin | 1.18 | 1.00 | 0.75 | 1.08 | 1.21 |
| | Whole basin | 32.53 | 32.45 | 32.24 | 32.33 | 32.44 |

seriously exceeded the standard, and the inferior V class accounted for about 90%. The water quality of the lake center was still in Class I.

## LULC variation characteristics

The LULC information classification results can be used to further analyze the LULC variation characteristics of the whole basin scale, village farmland areas, phosphate mine areas, and urban watershed scales in different pollution source areas (Table 2).

It can be seen from Table 2 that under the whole basin scale, woodland and grass cover, water, and cultivated land are the main LULC types in the basin (above 95% of total area), while other land accounts for approximately 2%. The area of ISA and other land use has
generally increased and other land use has increased by nearly five times during the period starting in 2005. The coverage of forest and grass and planting land has decreased but the area of water has remained basically unchanged. The significant characteristic of LUCC can be summarized as follows: the LULC type that is closely related to human economic activity shows strong growth due to economic development and the increase in human activities.

Forest and grass cover, and crop land are the dominant LULC types in mountainous watersheds from 2005 to 2017. Their area accounts for about 65% and 27% of the mountainous watershed, followed by impervious surfaces and waters, which accounts for 6% and 1% of the mountainous watershed, and the smallest is other land, accounting for only about 0.3%. Due to the economic development of the basin and the increase of human activities, a large number of artificial digs, deserts, and bare grounds have emerged resulting in an overall increase of the area of impervious surfaces and other land uses. Since implementing the policy of returning farmland to forests in 2003, the area of planted land has decreased. While this is related to the planting structures and patterns mainly based on economic crops (greenhouse vegetables, flowerbeds, nurseries, economic forests, etc.). The coverage of forest and grass has decreased slightly due to the artificial destruction of forest land in the mountainous areas natural succession, and land degradation, however, the area of water has remained basically unchanged.

From 2005 to 2017 in the phosphate mine basin, forestry and grass cover and cropping land are also the dominant LULC types. Their area accounts for 57% and 33% of the phosphate mine watershed, respectively, followed by the impervious surface and other land use, which accounts for about 5% and 3% of the phosphate mine basin, and the water area only accounts for about 1.4%. The phosphate mine has been completely banned in the Fuxian Lake basin; however, the mining area of the phosphate mine has left a variety of hidden dangers to the region, which has a greater impact on soil and crops, resulting in a reduction in the area of planted land. The implementation of a comprehensive ban on phosphate mining has resulted in a large number of abandoned factories, phosphate deposits, artificially excavated land, and naturally degraded deserts and exposed surfaces. However, the implementation of the ecological restoration project in the phosphate mine area has shown an increasing trend in the impervious surface, other land use, and forest -grass cover, and the area of water has remained basically unchanged.

The forestry and grass cover, crop land, and impervious surface are the dominant LULC types in the urban watersheds from 2005 to 2017. Their area accounts for 48%, 30%, and 20% of the urban watershed, respectively, while other land and water areas account for only 1.9% and 1.1%, respectively. An increase in urbanization has led to the implementation of a large number of land development projects in real estate, public facilities, and roads in the urban area which has occupied a certain amount of planted land and forest and grass cover, creating a greater area of hardened surface, building areas, structures and hardened road. These are accompanied by a great number of construction projects or long-term shutdowns, resulting in a large number of artificial digs or deserts and bare grounds. As a result, the area of planted land and the coverage of the forest and grass are generally

**Table 3  Correlation between LULC type and water quality.**

| Scale | Water quality index | CL | FGL | ISA | Othl | Water |
|---|---|---|---|---|---|---|
| Whole basin | COD | −0.794 | 0.272 | 0.612 | 0.901* | −0.008 |
| | TP | 0.941* | −0.532 | 0.769 | 0.933* | −0.327 |
| | TN | −0.444 | −0.159 | 0.722 | 0.491 | −0.228 |
| urban | COD | −0.994** | −0.949* | 0.975** | 0.823 | 0.255 |
| | TP | −0.473 | −0.692 | 0.405 | 0.048 | −0.558 |
| | TN | −0.187 | −0.543 | 0.174 | −0.087 | 0.106 |
| phosphate mine basin | COD | 0.532 | −0.118 | 0.620 | 0.848** | −.0275 |
| | TP | 0.924** | −0.406 | 0.925** | 0.821** | −.0622 |
| | TN | 0.477 | 0.078 | 0.538 | 0.427 | −.0158 |
| Village and farmland basin | COD | −0.784 | 0.626 | 0.753 | 0.774 | −0.093 |
| | TP | −0.955* | −0.922* | 0.892* | 0.810 | −0.403 |
| | TN | −0.986** | −0.966** | 0.929* | 0.797 | −0.600 |

**Notes.**

** indicates Sig. is a significant correlation at the 0.01 level (dual); * indicates Sig. is a significant correlation at the 0.05 level (dual).

decreasing and the impervious surface and other land use are increasing, while the area of water has remained basically unchanged.

## Multi-scale relationship analysis between LULC type and water quality

Correlation analysis was carried out on the water quality of Fuxian Lake and the proportion of LULC type area by the SPSS correlation analysis method. The results are shown in Table 3.

It can be seen from Table 3 that COD is closely related to other land, cropping land, and impervious surface at the whole basin scale. Among them, COD is significantly positively correlated with other land use and positively correlated with impervious surface but negatively correlated with cropping land. TP was significantly positively correlated with cropping land and other land use and positively correlated with impervious surface. However, there was a moderate negative correlation with forestry and grass cover and water body, but the significance was low. TN has no correlation with other LULC types except for a moderately positive correlation with impervious surfaces.

In the urban basin, COD was closely related to cropping land, forest cover, and impervious surfaces. Among them, cropping land was significantly negatively correlated with COD. TP was moderately negatively correlated with forestry and grass cover, and water body, but the significance was low; it was negatively correlated with cropping land, and had a low positive correlation with impervious surfaces. With artificial land there was no correlation with artificial digs, deserts, and bare ground. TN had no correlation with other land types except for a moderately negative correlation with forestry and grass cover.

In the phosphate mine basin, the phosphate mining area after reclamation was mostly in the LULC states of artificial excavation, bare, and cropping land. Mining areas that have not been reclaimed and mined are mainly classified as artificially excavated land and exposed ground. This feature is better reflected in the analysis results in Table 3. The values of COD

and TP were highly positively correlated with other land types (ie, artificially excavated land, desert, and bare ground), and TP was also highly positively correlated with cropping land and impervious surface.

In the village and farmland watershed, COD is moderately positively correlated with impervious surfaces and other sites (ie, artificially excavated land, desert, and bare ground), and was moderately negatively correlated with cropping land and forest, and grass cover, but the significance was low. TP was significantly negatively correlated with cropping land and was highly correlated with both forestry and grass cover, and impervious surface, but the forestry and grass cover was inhibited by TP, while the impervious surface was the opposite. Other land and TP were also positively correlated, but less significant. TN was significantly correlated with cropping land and forest cover with sig. values of 0.01. The impervious surface was highly positively correlated with TP, and the other land was moderately correlated with TP.

In summary, the effects of cropping land, forest cover, and impervious surface on TP and TN are significant. The reduction of cropping land did not improve the water quality, while forest and grass cover can effectively improve and purify water quality. Increases in artificial excavation, desert, and bare surface led to the deterioration of water quality, but the sensitivity between the two is low.

## Relationship model between LULC type and water quality in watershed scale

Based on the results of correlation analysis, using the significant variable $p < 0.05$ as the constraint, the stepwise entry-removal method was used in the SPSS software to establish the water quality and LULC type multiple regression equations at the sub-basin scale (Table 4).

From the regression model in Table 4, it can be seen that the COD of the Maliao River Basin has the strongest response to the cropping land, while the other water quality indicators and the LULC type response are not significant and the prepared significant $p$ value standard was not achieved. This is consistent with the results of the correlation analysis. In the phosphate mine basin, TP has the best fit to regression models of other land types and impervious surfaces, which further confirms that phosphate mining sites, bare ground surfaces, and impervious surfaces are one of the important factors in the increase of TP in this region. Through regression models, it was found that cropping land is one of the important indicators that led to higher TP and TN values in the village farmland.

In the whole basin, COD has the strongest response to other land use, while TP has the strongest correlation with cropping land. The TN index and LULC type response are not significant, and the prepared significant $p$-value standard was not achieved, which is consistent with the correlation analysis. It shows that the COD of the whole basin has the best fitting degree with the regression model of other land types, which further confirms that the basin phosphate mining point, bare surface, and artificial dig site are one of the important factors leading to the increase of COD value. Through the regression model, it was found that the proportion of planted land has a higher interpretation to TP,

**Table 4  Relationship model between water quality of entering lake river and LULC type at sub-basin scale.**

| Scale | Water quality index | Regression equation | $R^2$ | Adjusted $R^2$ | Sig. |
|---|---|---|---|---|---|
| | COD | COD = 99.199–1.733 FarmL | 0.988 | 0.984 | 0.001 |
| Urban basin | TP | Variable removed | – | – | – |
| | TN | Variable removed | – | – | – |
| | COD | COD =−19.68 + 12.27 OthL | 0.719 | 0.684 | 0.002 |
| phosphate mine basin | TP | TP = −0.44 + 0.089ISA + 0.061 OthL | 0.950 | 0.936 | 0.000 |
| | TN | Variable removed | – | – | – |
| | COD | Variable removed | – | – | – |
| Village and farmland basin | TP | TP = 2.525−0.104 FarmL | 0.912 | 0.883 | 0.11 |
| | TN | TN = 78.632−3.171 FarmL | 0.973 | 0.964 | 0.02 |
| | COD | COD = 0.857 + 0.901 OthL | 0.811 | 0.749 | 0.037 |
| Whole basin | TP | TP = −0.057 + 0.941 FarmL | 0.886 | 0.848 | 0.017 |
| | TN | Variable removed | – | – | – |

**Notes.**

FarmL stands for cropping land, OthL stands for other land, and ISA stands for impervious surface.

which mainly indicates that the area of planted land determines the application amount of chemical fertilizers and pesticides, and thus affects the TP concentration.

## Relationship between LULC type and water quality of rivers entering the lake at buffer scale

The riparian zone serves as a transitional zone for the exchange of matter and energy between terrestrial and river ecosystems. The special geographical location determines its important functions and research value. The LULC type, structure, and function of the riparian zone will have a series of effects on the river water quality (*Liu et al., 2016*). Based on the nine main rivers in the different pollution source areas of the basin, the river buffer zones are generated at 100 m, 300 m, 500 m, 700 m, and 1,000 m. The relationship between the water quality and LULC types is explored and a relationship model based on the statistical analysis of the LULC structure of each buffer zone was built.

Correlation analysis was carried out on the water quality index of the rivers in the urban area, phosphate mine area, and village farmland, and the corresponding LULC types of the five riparian buffer zones. The results are shown in Table 5.

At the riparian buffer scale in the urban area, different LULC types have different effects on various water quality indicators, while the correlation between the LULC type and COD value is relatively close. Cropping land, forest cover, and water quality are negatively correlated. The impervious surfaces and other land uses are positively correlated with water quality, and the correlation between water and water quality is low. It shows that the impervious surface and other land use on this scale are one of the important factors leading to the deterioration of water quality. From the spatial scale, the significance of cropping land and COD is: 500 m>300 m>700 m>1,000 m>100 m. Except for the 100 m scale, the significance level is 0.05, and the other scales are 0.01. TP and TN also have a negative correlation with cropping land, but the correlation coefficient is much smaller than COD. There is a negative correlation between the forest and grass cover and water

Li et al. (2019), *PeerJ*, DOI 10.7717/peerj.7283

**Table 5  Correlation between water quality and LULC in different pollution source areas.**

| Scale | LULC type | Riparian buffer (meters) | COD (mg/L) | TP (mg/L) | TN (mg/L) |
|---|---|---|---|---|---|
| | | 100 | −0.939 * | −0.491 | −0.539 |
| | | 300 | −0.992 ** | −0.631 | −0.314 |
| | CL | 500 | −0.999 ** | −0.538 | −0.292 |
| | | 700 | −0.990 ** | −0.498 | −0.327 |
| | | 1000 | −0.978 ** | −0.466 | −0.320 |
| | | 100 | −0.816 | −0.288 | −0.625 |
| | | 300 | −0.703 | −0.258 | −0.748 |
| | FGL | 500 | −0.717 | −0.371 | −0.798 |
| | | 700 | −0.775 | −0.437 | −0.771 |
| | | 1000 | −0.798 | −0.419 | −0.720 |
| | | 100 | −0.341 | 0.339 | −0.372 |
| | | 300 | 0.277 | 0.757 | −0.084 |
| Urban basin | ISA | 500 | 0.463 | 0.849 | 0.085 |
| | | 700 | 0.618 | 0.914 * | 0.197 |
| | | 1000 | 0.827 | 0.877 | 0.203 |
| | | 100 | 0.839 | 0.171 | 0.449 |
| | | 300 | 0.894 * | 0.192 | 0.107 |
| | Othl | 500 | 0.854 | 0.107 | −0.008 |
| | | 700 | 0.849 | 0.094 | −0.028 |
| | | 1000 | 0.862 | 0.109 | 0.052 |
| | | 100 | 0.393 | −0.414 | 0.701 |
| | | 300 | 0.283 | −0.486 | 0.192 |
| | Water | 500 | 0.414 | −0.402 | 0.114 |
| | | 700 | 0.497 | −0.389 | 0.075 |
| | | 1000 | 0.545 | −0.343 | −0.018 |

Li et al. (2019), *PeerJ*, DOI 10.7717/peerj.7283

**Table 5** (*continued*)

| Scale | LULC type | Riparian buffer (meters) | COD (mg/L) | TP (mg/L) | TN (mg/L) |
|---|---|---|---|---|---|
| | | 100 | −0.443 | −0.777** | −0.531 |
| | | 300 | −0.573 | −0.882** | −0.643* |
| | CL | 500 | −0.612 | −0.871** | −0.625 |
| | | 700 | −0.572 | −0.755* | −0.579 |
| | | 1000 | −0.434 | −0.499 | −0.477 |
| | | 100 | −0.283 | −0.665* | −0.327 |
| | | 300 | −0.267 | −0.718* | −0.378 |
| | FGL | 500 | −0.303 | −0.727* | −0.519 |
| | | 700 | −0.315 | −0.670* | −0.497 |
| | | 1000 | −0.389 | −0.689* | −0.472 |
| | | 100 | 0.109 | 0.549 | 0.324 |
| | | 300 | 0.399 | 0.759* | 0.605 |
| phosphate mine basin | ISA | 500 | 0.522 | 0.863** | 0.617 |
| | | 700 | 0.629 | 0.874** | 0.754* |
| | | 1000 | 0.584 | 0.650* | 0.788** |
| | | 100 | 0.863** | 0.734* | 0.643* |
| | | 300 | 0.842** | 0.787** | 0.417 |
| | Othl | 500 | 0.843** | 0.939** | 0.363 |
| | | 700 | 0.814** | 0.958** | 0.333 |
| | | 1000 | 0.828** | 0.919** | 0.393 |
| | | 100 | −0.313 | −0.658* | −0.203 |
| | | 300 | −0.269 | −0.573 | −0.078 |
| | Water | 500 | −0.278 | −0.585 | −0.086 |
| | | 700 | −0.304 | −0.607 | −0.099 |
| | | 1000 | −0.324 | −0.631 | −0.149 |

**Table 5 (*continued*)**

| Scale | LULC type | Riparian buffer (meters) | COD (mg/L) | TP (mg/L) | TN (mg/L) |
|---|---|---|---|---|---|
| | | 100 | −0.365 | −0.767 | −0.886* |
| | | 300 | −0.590 | −0.891* | −0.976* |
| | CL | 500 | −0.600 | −0.902* | −0.979* |
| | | 700 | −0.656 | −0.933* | −0.990* |
| | | 1000 | −0.659 | −0.924* | −0.988* |
| | | 100 | −0.095 | −0.371 | −0.543 |
| | | 300 | −0.198 | −0.589 | −0.724 |
| | FGL | 500 | −0.303 | −0.684 | −0.794 |
| | | 700 | −0.369 | −0.711 | −0.805 |
| | | 1000 | −0.392 | −0.688 | −0.760 |
| | | 100 | 0.551 | 0.794 | 0.883* |
| Village and farmland basin | | 300 | 0.666 | 0.828 | 0.909* |
| | ISA | 500 | 0.682 | 0.828 | 0.906* |
| | | 700 | 0.736 | 0.875 | 0.925* |
| | | 1000 | 0.623 | 0.783 | 0.876 |
| | | 100 | 0.830 | 0.601 | 0.475 |
| | | 300 | 0.824 | 0.659 | 0.554 |
| | Othl | 500 | 0.760 | 0.697 | 0.623 |
| | | 700 | 0.676 | 0.668 | 0.592 |
| | | 1000 | 0.781 | 0.862 | 0.804 |
| | | 100 | 0.689 | 0.719 | 0.596 |
| | | 300 | 0.526 | 0.727 | 0.661 |
| | Water | 500 | 0.576 | 0.700 | 0.605 |
| | | 700 | 0.619 | 0.716 | 0.614 |
| | | 1000 | 0.548 | 0.661 | 0.570 |

quality, indicating that forest and grass cover has a filtering effect on pollutants and can reduce water pollution. Forest and grass cover have the greatest impact on COD, followed by TN, and TP. Except for individual scales, the impervious surface is basically positively correlated with water quality, and the relationship between COD and TP is most significant on the scale of 500 m–1,000 m. Other land types are highly positively correlated with COD and are highly sensitive at 300 m. However, there is basically no correlation between other land use and TP and TN, indicating that other sites have little or no effect on TP and TN.

At the riparian buffer scale in the phosphorus area, a more significant feature reveals that other LULC types have a higher correlation with TP except for the lower correlation of waters. COD and TN are weakly correlated or even not related to the other four types of land use except for the correlation with artificial landfill, desert, and bare ground. The TP and LULC types in the phosphate mine zone show a high correlation and sensitivity. This fully demonstrates that densely distributed phosphate deposit sites and phosphorus chemical companies are the main sources of phosphorus pollution in the region.

Compared with the urban scale and the phosphate mine scale, the sensitivity between water quality indicators and the LULC type is significant in village farmland. The LULC type has a significant correlation with TN. There was a negative correlation between cropping land and COD, TP, and TN among which TN was the most significant, TP was the second, and COD was the lowest. The degree of correlation with TN is somewhat different at each buffer scale. The order of significance is: 700 m>1,000 m>500 m>300 m>100 m, which is basically consistent with TP. The degree of correlation with COD is low, with a moderate negative correlation between 500 m and 1,000 m, and a low negative correlation between 100 m and 300 m.

There was a moderate negative correlation between forest and grass cover and TN and TP. It was negatively correlated with COD at 500 m–1,000 m, and the correlation between 100 m and 300 m was not obvious.

The degree of impervious surface and water quality correlation was: TN>TP>COD. This law is exactly the opposite of that for other land use and water quality. The correlation between other land (artificial dig, desert and bare ground) and water quality is COD>TP>TN. This phenomenon indicates the impervious surface and other land use play different roles in the increase of water quality indicators in the region. The impervious surface mainly leads to the increase of the TN index, and other land use leads to the increase of the COD index. From the buffer scale in the 700 m buffer zone, the correlation between the impervious surface and water quality is still the highest, followed by 300 m, while the correlation between other land and water quality has a certain degree of differentiation, and TN and TP basically have 1,000 m–100 m in a gradually decreasing trend, which is exactly the opposite of COD. The proportion of the water area is basically within 4%, and the correlation with water quality is small.

## Relationship model between river water quality and LULC under buffer scale

The multivariate regression equation of water quality and the LULC type at the riparian buffer scale is constructed by the SER method in the different pollution source areas (Table 6).

Table 6 shows that the LULC types in different regions reveal significant differences on water quality indicators. The urban area has a greater impact on COD and the phosphate mine area has a greater impact on TP. The village farmland area has a greater impact on TN. The impact of the urban LULC type on COD is: 300 m>500 m>sub-basin>700 m>1,000 m>100 m>whole basin; the effect of the LULC type on COD in the phosphate mine area is: 300 m>500 m>whole basin>100 m>sub-basin>1,000 m>700 m; the influence of the LULC type on TP in the phosphate mine area is: 700 m>500 m>sub-basin>300 m>whole basin>1,000 m>100 m, the influence of the LULC type on TN in village farmland is: 700 m>1,000 m>sub-basin >500 m>300 m>100 m. The impact of the LULC type on TP in village farmland is: sub-basin>full basin 700 m>1,000 m>500 m>300 m>100 m. From the sorting point of view, the urban areas and phosphate mines have a greater impact on COD and TP at 300 m, 500 m, and 700 m, while in the village farmland, the sub-basin scale has a greater impact on water quality.

## DISCUSSION

The percentage of area of ISA and other land (man-made land, desert, and bare ground) is positively correlated with water quality, regardless of whether it is in the whole basin or buffer zone of different pollution source areas, which has a positive effect and is consistent with relevant research conclusions (*Osborne & Wiley, 1988*; *Basnyat et al., 1999*; *Sliva & Williams, 2001*; *Guang et al., 2008*; *Huang et al., 2011*; *Yang et al., 2017*; *Xiang et al., 2018*). ISA and other land use ratios are positively correlated with COD and TP. This is mainly due to the fact that the amount of organic matter and phosphorus loss is closely related to the content and concentration of runoff. The acceleration of urbanization has led to an increase in the area of artificial buildings and the surface area of impervious surfaces has increased accordingly. Pollutants such as roads, plazas, and roofs are more likely to converge and carry large amounts of sediment, nutrients, and heavy metal contaminants, which easily flow into the river, creating a burden on the river. In addition, the discharge of industrial and domestic wastewater from urban areas into water bodies can also cause the water quality to decline (*Fedorko et al., 2005*).

Forest and grass covers are another important factor affecting water quality and are negatively correlated with water quality. This is consistent with the expected understanding that forest and grass cover can improve water quality. Forest and grass cover can reduce runoff, thereby reducing soil erosion and water quality degradation due to soil erosion. The results show that most of the water quality parameters that characterize the water quality decline are negatively correlated with the percentage of forest cover area, which is consistent with other research conclusions. The vegetation as a "sink" of pollutants confirms that vegetation has the effect of intercepting and purifying pollutants (*Osborne*

**Table 6  Multi-scale relationship model of LULC type for water quality in different pollution source areas.**

| Spatial scale | buffer | Water quality index | Regression equation | $R^2$ | Adjusted $R^2$ | Sig. |
|---|---|---|---|---|---|---|
| urban | | COD | COD = 80.589–1.562 FarmL | 0.881 | 0.841 | 0.018 |
| | 100 m | TP | Variable removed | – | – | – |
| | | TN | Variable removed | – | – | – |
| | | COD | COD = 96.942–1.453 FarmL + 0.712 OthL | 1.0 | 1.0 | 0.000 |
| | 300 m | TP | Variable removed | – | – | – |
| | | TN | Variable removed | – | – | – |
| | | COD | COD = 96.819–1.677 FarmL | 0.999 | 0.998 | 0.000 |
| | 500 m | TP | Variable removed | – | – | – |
| | | TN | Variable removed | – | – | – |
| | | COD | COD = 91.237–1.505 FarmL | 0.981 | 0.975 | 0.001 |
| | 700 m | TP | TP = −1.657 + 0.088 ISA | 0.835 | 0.780 | 0.030 |
| | | TN | Variable removed | – | – | – |
| | | COD | COD = 89.959–1.485 FarmL | 0.957 | 0.942 | 0.004 |
| | 1000 m | TP | Variable removed | – | – | – |
| | | TN | Variable removed | – | – | – |
| Phosphate mining area | | COD | COD = −0.201 + 10.676 othL | 0.744 | 0.712 | 0.001 |
| | 100 m | TP | TP = 1.372–0.029 FarmL | 0.604 | 0.554 | 0.08 |
| | | TN | TN = 3.089 + 1.167 othL | 0.413 | 0.340 | 0.45 |
| | | COD | COD = −29.907 + 11.726 othL + 4.495 Water | 0.878 | 0.843 | 0.01 |
| | 300 m | TP | TP = 1.118–0.029 FarmL + 0.043 othL | 0.893 | 0.862 | 0.00 |
| | | TN | TN = 18.767–0.395 FarmL | 0.414 | 0.341 | 0.045 |
| | | COD | COD = −29.745 + 11.223 othL + 5.198 Water | 0.860 | 0.820 | 0.001 |
| | 500 m | TP | TP = 0.844 + 0.065 othL–0.021 FarmL | 0.990 | 0.987 | 0.00 |
| | | TN | Variable removed | – | – | – |
| | | COD | COD = −2.252 + 8.166 othL | 0.663 | 0.621 | 0.04 |
| | 700 m | TP | TP = −0.542 + 0.073 othL + 0.8 ISA | 0.997 | 0.996 | 0.00 |
| | | TN | TN = −10.207 + 2.091 ISA | 0.568 | 0.514 | 0.012 |
| | | COD | COD = −10.61 + 12.501 othL | 0.685 | 0.646 | 0.003 |
| | 1000 m | TP | TP = −0.11 + 0.148 othL | 0.845 | 0.825 | 0.000 |
| | | TN | TN = −19.188 + 3.269 ISA | 0.621 | 0.573 | 0.07 |
| Village and farmland | | COD | Variable removed | – | – | – |
| | 100 m | TP | Variable removed | – | – | – |
| | | TN | TN = 55.895–1.032 FarmL | 0.785 | 0.713 | 0.045 |
| | | COD | Variable removed | – | – | – |
| | 300 m | TP | TP = 2.515–0.057 FarmL | 0.794 | 0.725 | 0.042 |
| | | TN | TN = 79.528–1.863 FarmL | 0.953 | 0.937 | 0.004 |
| | | COD | Variable removed | – | – | – |
| | 500 m | TP | TP = 2.46–0.06 FarmL | 0.813 | 0.751 | 0.036 |
| | | TN | TN = 77.098–1.938 FarmL | 0.958 | 0.944 | 0.004 |
| | | COD | Variable removed | – | – | – |
| | 700 m | TP | TP = 2.638–0.067 FarmL | 0.870 | 0.827 | 0.021 |
| | | TN | TN = 81.143–2.11 FarmL | 0.980 | 0.973 | 0.001 |
| | | COD | Variable removed | – | – | – |
| | 1000 m | TP | TP = 2.768–0.073 FarmL | 0.854 | 0.805 | 0.025 |
| | | TN | TN = 85.869–2.319 FarmL | 0.977 | 0.969 | 0.001 |
*& Wiley, 1988*; *Sliva & Williams, 2001*; *Novotny, 2002*; *Bahar, Ohmori & Yamamuro, 2008*; *Lopez et al., 2008*; *Zeng et al., 2012*; *Putro et al., 2016*).

Cropping land is not a major factor in explaining the decline in water quality. At the whole basin scale, TP has the strongest correlation with cropping land, and the TN index is not significant. The impact of cropping land on water quality is complex and negatively correlated with TN and TP. This is consistent with the conclusion that the proportion of agricultural land is negatively correlated with TN (*Lenat & Crawford, 1994*; *Johnson et al., 1997*; *Sliva & Williams, 2001*; *Chang, 2008*; *Tu & Xia, 2008*). However, it is also different from some studies (*Fedorko et al., 2005*; *Bahar, Ohmori & Yamamuro, 2008*). This indicates that the area ratio of agricultural land is not the main factor affecting TN, which is related to the application amount of pesticides and fertilizers, planting structure, and the distance from the receiving water body and topography. At the same time, the cropping land contributes a large amount of nutrient salt to the river through fertilization with farmland runoff, yet it also acts as a vegetation or wetland system to attach, absorb, and retain pollutants. From the spatial distribution of LULC in the Fuxian Lake basin, the arable land in the Fuxian Lake Basin is mainly distributed throughout the mountainous areas and the flat area on the north bank of Fuxian Lake. The river that enters the lake passes through this arable area and then flows into Fuxian Lake. The arable land in the flat area has a certain retention and retention effect on the TN of the river into the lake and is related to the natural characteristics of TN and its migration and transformation (*Peng et al., 2015*). A large number of microorganisms in the arable land are beneficial to nitrification and denitrification between the different forms of nitrogen (*Zeng et al., 2012*; *Li et al., 2016*). However, this is related to the policies and implementation of the prevention and control of non-point source pollution in the Fuxian Lake Basin by the Yunnan Provincial Government and the Yuxi Municipal Government in recent years. These policies and measures limit the use of ammonia and phosphate fertilizers and pesticides in the basin, resulting in a negative correlation between TN, TP, and planted land.

From the perspective of the spatial effects of the LULC types and water quality, there are significant spatial scale differences in the relationship between the LULC types and water quality, and the spatial scale differences in various regions are unique. Most studies have shown that LULC has the highest interpretation rate of water quality at the small watershed scale and it is the spatial scale with the strongest impact on the water quality of the river (*Sliva & Williams, 2001*; *Tudesque, Tisseuil & Lek, 2014*; *Ding et al., 2016*). However, some studies have found that the relationship between LULC and water quality in the 100 m buffer zone is more significant than the whole basin scale (*Shen et al., 2012*; *Zhou et al., 2016*). Those results are different from the results of this study and are related to the LULC characteristics of the study area itself. At the sub-basin scale, forest and grass cover is the dominant land type, accounting for more than 50% of the area, much higher than other land types. The superior land use type is prominent, which consistently creates a higher interpretation rate of its relationship with water quality. Compared with the riparian buffer scale, cropping land, ISA, forest and grass cover together constitute a dominant land type, and the land type components are complex and the effects of various types on water quality are different.

The existence of superposition or "waxing and waning" results in a lower interpretation rate of water quality than the sub-basin scale (*Chang, 2008*; *Tudesque, Tisseuil & Lek, 2014*; *Yu et al., 2015*; *Xiang et al., 2018*). The results show that the 300 m riparian buffer zone in the urban watershed is the strongest spatial scale for the LULC type to affect COD. This is consistent with the results of research conducted by relevant scholars in Aibi Lake in Xinjiang, Erhai and Fuxian Lake in Yunnan (*Liu, Wu & Gao, 2008*; *Cao et al., 2018*; *Xiang et al., 2018*). It can be inferred that for the plateau lake basin, the 300 m riparian buffer is the strongest spatial scale for the LULC type to affect COD. This inference needs further verification. However, due to the different contributions of ISA and cropping land to the generation and emission of pollutants, the influence of different spatial patterns on the pollution of river waters presents a "waxing and waning" relationship. Therefore, it is necessary to properly plan the proportion of land types in the riparian zone and control the development of urban land in the river basin.

## CONCLUSIONS

The quality of water entering the lake is generally poor, while the quality of water in the lake center is still type I. The annual average water quality of the rivers entering the lake is inferior to the V category. Chemical Oxygen Demand (COD) in 45 sections had the highest proportion of type I and type II, reaching 53.33%, the other types of Total Phosphorus (TP) were evenly distributed, accounting for 22–26%. Total Nitrogen (TN) seriously exceeded the standard, revealing that nearly 90% of the water quality is inferior V. The spatial variation of the water quality of the rivers entering the lake is significant. The urban watershed has a higher COD value, the phosphate mine watershed has a higher TP value, and the mountain watershed has a higher TN value. The changes of COD, TN, and TP generally revealed a growing trend in each sub-basin, and the variation was significant.

The spatial variation of LUCC in the basin is significant. With an economy focused on development and the enhancement of human activities, as well as the increase of urbanization in the basin, the cultivated land in sub-basins of urban areas, phosphate mines, and village farmland shows a decreasing trend, while ISA and other land use are increasing, and the water area basically remains the same. The woodland and grass in the phosphate basin is increasing, while it is decreasing in urban areas and villages.

Multi-scale relationships exist between the LULC types and water quality. In the analysis of the relationship between the LULC type and water quality, the spatial heterogeneity of LULC and pollution sources in the basin were fully believed to affect the water quality indicators, and the multi-scale relationship analysis between LULC and water quality from different pollution sources within the basin was strengthened. This enhances some of the research uncertainty and highlights the multi-scale relationship between LULC changes and the water quality indicators. There is a significant difference in the interpretation of the relationship between water quality and LULC types and patterns. The area percentage of ISA and artificial excavation land, desert, and bare surface plays a promoting role in each spatiotemporal scales, while planting land and forest grass cover plays an inhibitory role. Forest cover can be expected to improve the water quality. There is no obvious correlation

between the increase or decrease of planting land and water quality. The impact of planting land on water quality is complex and negatively correlated with TN and TP. This indicates that the area percentage of agricultural land is not the main factor affecting TN, but is related to the application of pesticides and fertilizers, planting structure, and distance from the receiving water body and topography.

The multi-scale relationship between the LULC types and water quality is significant. The 300 m riparian buffer zone in urban watersheds is the strongest spatial scale (feature scale) of the LULC type effect on COD. This is consistent with the results of other plateau lake studies, and the universality of this conclusion needs further verification. There are similarities between the effect the LULC pattern has on the water quality and the effect of the LULC type on the water quality index. The urban LULC type and pattern have the greatest impact on COD, the phosphate mine area has the greatest impact on TP, and the village farmland has the greatest impact on TN. The difference is as follows: urban and phosphate mines have a higher interpretation of COD and TP in the buffer zone of 300 m, 500 m, and 700 m, while in the village farmland, the sub-basin scale has the highest interpretation of water quality. The influence of the LULC pattern on the water quality indicators in different regions are as follows: in urban areas and phosphate mines, COD and TP are better explained in sub-basin; in village farmland, TN can be better explained in 700 m buffer zone, and TP has a higher interpretation in the buffer zone at 300 m.

The regional differences in the watershed environmental protection measures are significant from the perspective of preventing contamination, improving the quality of water entering the basin, and referencing the water quality threshold landscape components. In the village farmland area, the ISA ratio should be reduced from the current 6% to less than 5%. The adjustment should be most effective in the 700-meter river bank and correspondingly reduce the area of other land within the riparian buffer at 1,000 m, especially artificial digs, deserts, and bare ground. The coverage of forest and grass in the range of 500 m to 1000 m on the riparian buffer should also be increased as should a focus on protecting the existing types of forest and grass cover.

In the phosphate mining area, the area of other LULC types (artificial excavation, desert, and bare ground) should be reduced. At the same time, the proportion of ISA should be decreased to less than 5%, reducing the area of planted land and increasing the coverage of forest and grass, especially to strengthen the restoration of forest and grass cover on the phosphate rock land and the land use remediation after the relocation of phosphorus chemical enterprises. According to the correlation between the land bank and the water quality in the river bank in the 100-meter buffer zone of the river bank, it is necessary to focus on the restoration of forest and grass cover on the phosphate rock land (desert and bare ground), as well as the large amount of labor required after the relocation of the phosphorus chemical enterprise. The treatment of landfill and a restoration of surface cover should also be addressed. In the buffer zone of 500 m to 700 m, the proportion of ISA is mainly reduced, and the proportion of forest and grass cover is increased at 300 to 500 m.

There is a need to refer to the results of previous studies for the urban areas (*Theobald et al., 2009*; *Liu, Wu & Gao, 2008*; *Liu, Li & Peng, 2010*). The main city area is at a threshold
level of degradation (48%). Therefore, the urban area should greatly control the proportion of ISA, minimize the proportion of ISA, and increase the coverage of forest and grass. Although other land types (artificial excavation, desert and bare ground) are not related to water quality indicators, recovery also makes sense. From the sensitive scale of the river bank buffer zone, the adjustment of the above-mentioned land types is most effective in the range of 300 m from the river bank.

With the rapid development of the regional social economy and the acceleration of urbanization, the effects on the water environment caused by the change of LULC has become a vital academic issue to address. The understanding and correlation between LULC and water quality have more commonalities, but due to the combination of regional characteristics, scale effects, and data quality (second-hand data), the research results can be unique or lead to uncertainty. Due to many factors affecting lake basins and rivers, complex interactions, and limitations in research data, time, and author level, the mechanism of LULC changes on water quality, and the monthly and seasonal equivalents of LULC types, and water quality, a multi-time-scale response will be carried out at a later time.

### Funding

This work was supported by the National Natural Science Foundation of P.R. China (A coupling effects study of spatial and temporal process between impervious surface area pattern and soil erosion of Dianchi Lake Basin during fast urbanization, Grant no. 41561086; Future land use/cover simulation and its response mechanism to water quality by coupling human activities and climate change effects in Fuxian Lake watershed, Grant no. 41861051, Research on cloud resources auto scaling and load balancing by considering the complexity of spatiotemporal computing, Grant no. 41661086), and the National Geographical state monitoring demonstrative project —"Dynamic monitoring for ecological environment in Fuxian lake watershed" from the National Administration of Surveying, Mapping and Geoinformation (file no. 201435). The funders had no role in study design, data collection and analysis, decision to publish, or preparation of the manuscript.

### Grant Disclosures

The following grant information was disclosed by the authors:
National Natural Science Foundation of P.R. China: 41561086, 41861051.
Research on cloud resources auto scaling and load balancing: 41661086.
National Administration of Surveying, Mapping and Geoinformation: 201435.

### Competing Interests

The authors declare there are no competing interests.

### Author Contributions

- Shihua Li conceived and designed the experiments, authored or reviewed drafts of the paper.

- Shuangyun Peng performed the experiments, analyzed the data.
- Baoxuan Jin contributed reagents/materials/analysis tools.
- Junsong Zhou prepared figures and/or tables.
- YingXin Li approved the final draft.

## Data Availability

The raw data are available in Tables S1 to S6.

## Supplemental Information

Supplemental information for this article can be found online at http://dx.doi.org/10.7717/peerj.7283#supplemental-information.

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
