# Peer review of "Multi-scale relationship between land use/land cover types and water quality in different pollution source areas in Fuxian Lake Basin"

_PeerJ, doi:10.7717/peerj.7283_

## Round 0.1 · original submission · Major Revisions

Two reviewers have provided detailed comments on your manuscript with varying recommendations. Overall the topic of this study is important, however, both reviewers noted the obvious shortcomings. I would gave the authors a chance to revise if they can address all the issues carefully.

Reviewer 1 ·

Basic reporting

The basic information of this paper is ok

Experimental design

The expeimental design of this paper is reasonable

Validity of the findings

Some intersting and meaningful results have been found

Additional comments

Fuxian Lake is a famous plateau freshwater lake with good water quality and deep lake water.
It is important to carry out the attribution analysis of water quality change here. The authors should supply more details about the data, and more important, point out the novelty of this study and also strengthen related discussion.

L.15-17: Point out the research significance or scientific question of this study
L. 56-85: Add more relevant references in Yunnan or Fuxian Lake
L. 159-162: The basis or meaning of such scale division needs to be emphasized
L.187-192: The test method for water quality data, accuracy (year, month or day) needs to be explained, the full name of TP, TN, COD needs to be added
L.205-203, L.372-375, those parts should remove to the Materials & Methods section.

Language: the language should be polish by native English speakers.

Tables and figures: to many tables and figures, it is recommended to delete

·

Basic reporting

1. The use of English needs to be improved throughout the paper, both to ensure that the content can be understood by reviewers and readers, and to make the paper more professional and of archival journal quality. Some examples where the language could be improved include lines:
line 69-77, much too long a sentence with grammar mistakes
line 80-85, also the same
line 98, the water quality is one of the best nature lakes?
line 99-101, three sentences were put together
...
The authors should check the whole paper to correct the similar expressions.


2. The field background/context provided should be more sufficient.
The article should include sufficient introduction and background to demonstrate how the work fits into the broader field of knowledge. Relevant prior literature should be appropriately referenced.

3. Article structure were clearly arranged and raw data were shared.


4. The result part should be well modified and further summarized.

Experimental design

1. Original primary research is within Aims and Scope of the journal. Research question is well defined, relevant & meaningful.

2. Rigorous investigation is performed to a high technical & ethical standard.

3. Methods described are simple and insufficient.

Validity of the findings

This study also got some new findings in the scale relationship between LULC (Land Use/Land Cover, LULC) types.
Data is robust, statistically sound, & controlled.
Conclusions needs to be appropriately stated, should be connected to the original question investigated, and should be limited to those supported by the results.

Additional comments

Based on a variety of scales, this paper discusses the relationship between land use and water environment in different water source polluted areas in Fuxian lake basin. The research results are of great scientific significance for the rational planning of land type proportion in riparian zone, the control of urban land development and the improvement of water quality in the basin.

1. Why ignore training and samples of the 2005 in the study period. line 143

2. The whole research method is too simple. In the evaluation of the characteristics of land use, maybe it can be considered to construct a new characteristic index to re-discuss the relationship between land use and water quality.

3. It is suitable to choose three representative rivers polluted by phosphate rock, urban development and farmland as the analysis pair in the study. Give some more data support?

4. Many scholars may select the whole basin, sub-basin and buffer of three dimensions in order to develop relationship between land use and water quality of research. This paper chose a variety of scales to explore Fuxian lake watershed land use different water pollution and water quality, please show the main innovation points of this article?

5. In this paper, selection of the whole basin, basin and buffer three scales, among them, the buffer size to choose the 100 m and 300 m, 500 m, 700 m and 1000 m, 300 m buffer scales are discussed is the biggest impact of the spatial scale, but there is no contrast buffer 300 m scale and whole watershed scale, watershed scale effect on the relationship between land use and water quality.

6. The conclusion part should be more simple and clear to show the reader the what conclusion can be made more clearly.

7. If the decrease of cultivated land does not improve the water quality, can it be concluded that one of the important reasons for the deterioration of water quality in this area is the rapid increase of impervious surface? Is this conclusion representative or taken out of context?

---

## Round 0.2 · accepted · Accept

The authors have made great progress according to the reviewers' comments. Hence, I am pleased to tell you that your work has now been accepted for publication in PeerJ.

Reviewer 1 ·

Basic reporting

no comments

Experimental design

no comment

Validity of the findings

no comment

·

Basic reporting

Clear and unambiguous, professional English used throughout.

Experimental design

Original primary research within Aims and Scope of the journal

Validity of the findings

corrected well

Additional comments

The authors have corrected all the questions, and answered the in the correct way.